# Springs and Springs-Dependent Taxa of the Colorado River Basin, Southwestern North America: Geography, Ecology and Human Impacts

**Lawrence E. Stevens \***, **Jeffrey Jenness** and **Jeri D. Ledbetter**

Springs Stewardship Institute, Museum of Northern Arizona, 3101 N. Ft. Valley Rd., Flagstaff, AZ 86001, USA;
Jeff@SpringStewardship.org (J.J.); Jeri@SpringStewardship.org (J.D.L.)
**\*** Correspondence: Larry@SpringStewardship.org

**Abstract:** The Colorado River basin (CRB), the primary water source for southwestern North America, is divided into the 283,384 km$^2$, water-exporting Upper CRB (UCRB) in the Colorado Plateau geologic province, and the 344,440 km$^2$, water-receiving Lower CRB (LCRB) in the Basin and Range geologic province. Long-regarded as a snowmelt-fed river system, approximately half of the river's baseflow is derived from groundwater, much of it through springs. CRB springs are important for biota, culture, and the economy, but are highly threatened by a wide array of anthropogenic factors. We used existing literature, available databases, and field data to synthesize information on the distribution, ecohydrology, biodiversity, status, and potential socio-economic impacts of 20,872 reported CRB springs in relation to permanent stream distribution, human population growth, and climate change. CRB springs are patchily distributed, with highest density in montane and cliff-dominated landscapes. Mapping data quality is highly variable and many springs remain undocumented. Most CRB springs-influenced habitats are small, with a highly variable mean area of 2200 m$^2$, generating an estimated total springs habitat area of 45.4 km$^2$ (0.007% of the total CRB land area). Median discharge also is generally low and variable (0.10 L/s, $N = 1687$, 95% CI = 0.04 L/s), but ranges up to 1800 L/s. Water pH and conductivity is negatively related to elevation, with a stronger negative relationship in the UCRB compared to the LCRB. Natural springs water temperature and geochemistry throughout the CRB varies greatly among springs, but relatively little within springs, and depends on aquifer hydrogeology, elevation, and residence time. As the only state nearly entirely included within the CRB, Arizona is about equally divided between the two geologic provinces. Arizona springs produce approximately 0.6 km$^3$/year of water. Data on >330 CRB springs-dependent taxa (SDT) revealed at least 62 plant species; 216 aquatic and riparian Mollusca, Hemiptera, Coleoptera, and other invertebrate taxa; several herpetofanual species; and two-thirds of 35 CRB fish taxa. Springs vegetation structure, composition, and diversity vary strongly by springs type, and plant species density within springs is high in comparison with upland habitats. Plant species richness and density is negatively related to elevation below 2500 m. Human population in and adjacent to the CRB are growing rapidly, and ecological impairment of springs exceeds 70% in many landscapes, particularly in urbanized and rangeland areas. Anthropogenic stressors are primarily related to groundwater depletion and pollution, livestock management, flow abstraction, non-native species introduction, and recreation. Ensuring the ecological integrity and sustainability of CRB groundwater supplies and springs will require more thorough basic inventory, assessment, research, information management, and local ecosystem rehabilitation, as well as improved groundwater and springs conservation policy.

**Keywords:** biodiversity; Colorado River Basin; ecosystem ecology; groundwater; human population growth; species; springs; springs-dependent taxa

---

## 1. Introduction

Groundwater reaches and often flows from the Earth's surface, generating discrete, groundwater-dependent, surface-linked headwater aquatic-wetland-riparian ecosystems (here referred to as springs). Springs often are biologically and culturally complex, highly individualistic, strongly ecotonal, and ecologically highly interactive, functioning as "keystone ecosystems"—small patches of habitat that play ecologically influential roles in adjacent landscapes [1]. Lindeman's [2] trophic-dynamic concept of ecosystem ecology was ground-truthed at Silver Springs in Florida [3], but springs ecosystem ecology has, until recently, been generally neglected. Although widely recognized for their scientific, biological, and socio-cultural value and complexity, springs everywhere are heavily altered and appropriated (e.g., [4–8]). Both the ecological importance of springs as keystone ecosystems and the level of ecological impairment of springs is particularly great in arid regions [4], such as the Colorado River basin (CRB) in southwestern North America. Several subregional analyses of springs indicate that even in well-protected national park lands, more than half of the CRB springs inventoried have been ecologically impaired by human activities, and impairment levels commonly exceed 70% in less well-protected landscapes [4,8]. Societal alarm over the condition of desert springs is beginning to initiate proactive conservation responses. For example, threats to southwestern springs and the large, highly endemized truncalleloidean springsnail biodiversity they support has stimulated collaborative, proactive conservation planning by the states of Nevada and Utah in the USA [7,9]. Although springs are of great and growing scientific and conservation concern, comprehensive ecological assessment of springs distribution, characteristics, dependent species, connectivity to drainage networks, and conservation status remain outstanding in arid landscapes such as the CRB, where stewardship information and action are most urgently needed.

The arid CRB is 14% larger than the land area of France, and occupies parts of seven USA states and two northwestern Mexican states. CRB surface water hydrology has been intensively studied for the past century because as the region's dominant river—it is the primary water source and an important hydroelectric energy source for approximately 40 million people. The Colorado River flows through some of the world's most iconic landscapes, including Rocky Mountain, Canyonlands, and Grand Canyon National Parks [10–20]. The CRB also supports the highest percent of endemic ichthyofaunae of any North American river system [18], as well as many springs-dependent taxa (SDT; "crenobionts"), including endemic and rare plants, invertebrates, herpetofaunae, and other vertebrates. SDT are those organisms for which springs are essential for one or more life history stage, and/or for which the majority of the population occurs in springs-supported habitats. Long-regarded as a snowmelt-fed river system, the important contribution of groundwater to CRB flow and ecology has only recently been recognized. Springs, spring-fed lakes, and other groundwater-dependent ecosystems (GDEs) contribute approximately half of the Colorado River's total mean annual flow, including its critically important groundwater baseflow [11,19]. Virtually every natural perennial CRB tributary examined thus far is baseflow-fed by discrete springs, springfed wetlands, or small groundwater-fed lakes (e.g., [21]). Climate change is decreasing winter snow pack and infiltration, and is increasing CRB air temperature, evapotranspiration, and demand for groundwater [12,22]. Thus, the arid CRB is undergoing a climate change-induced transition [23,24], one in which the importance of sustainable groundwater is amplified, warranting focused attention on the water features that deliver the river's primary baseflow—its springs.

Here, we present a basin-wide analysis of CRB springs distribution, ecological characteristics, SDT, and the human influences and population growth issues acting on them. We review the existing literature, springs, and SDT databases (e.g., [25]), and use data from field inventories and specimen data compiled through museum data-mining in several southwestern university collections, the National Museum of Natural History, and the Museum of Northern Arizona. We examined geographic and ecohydrological variation in springs characteristics across elevation between the upper and lower halves of the CRB. We focus particular attention on the springs of Arizona, the nation's second driest state, because it is the only state that lies nearly entirely within the CRB, its springs and SDT have

been intensively studied, and it is nearly equally split between the two geologic provinces of the CRB. We further focus attention on springs in the Grand Canyon ecoregion, as well as the Verde and Virgin River basins, drainages that bridge the boundary between the two sub-basins. We discuss how improved information on CRB springs can improve ecological understanding of the roles of springs in this vast river basin, and how such understanding can contribute to enhanced natural resource stewardship at a critical time in CRB management history. We provide recommendations on CRB springs ecosystem management as the basin transitions towards a new, drier condition [23,24], one presently more strongly dominated by groundwater flow under changing climate conditions. Despite the present lack of CRB inter-state coordination in U.S. and international groundwater management, such information is essential to ensure future ground- and surface-water sustainability, and the many natural and human beneficiaries of those waters [25].

## 2. Methods

### Study Area

The Colorado River is the largest and most important river of southwestern North America, occupying a 627,824 km$^2$ aridlands basin (Figure 1). At 2330 km in length and with a 1906–2000 mean annual discharge of 16.3 km$^3$/year (range 6.5–29.6 km$^3$/year) [26,27], the CRB is bounded by the Rocky Mountains and the Continental Divide to the east, the Snake/Columbia River basin to the north, and the Basin and Range geological province boundary to the west and south. The CRB has been defined geologically, hydrologically, and politically. The hydrologic boundary of the CRB varies slightly from its geological boundaries by including Basin and Range White River drainage in Nevada, and excluding the endorheic Big Sandy drainage basin in southwestern Wyoming. Additional geographic information on the CRB is provided in the Supplementary Materials.

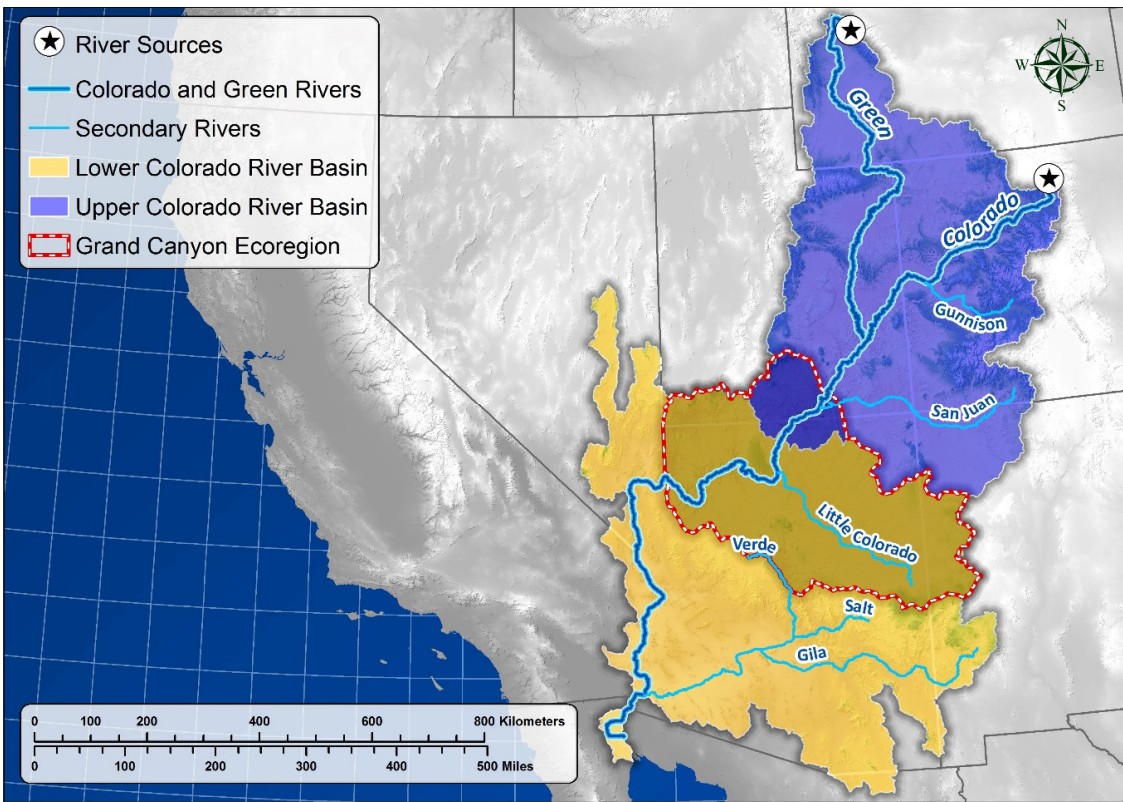

**Figure 1.** General map of the Colorado River Basin, depicting the Upper and Lower Basins, and the Grand Canyon ecoregion.

The CRB occupies two physiographic provinces, which naturally divide the upper from the lower basin. The 283,384 km$^2$ Upper Colorado River Basin (UCRB) drains the West Slope of the Rocky Mountains and the stratigraphically largely undeformed Colorado Plateau section of the Rocky Mountains geologic province. The UCRB ranges in elevation from 4365 m at Uncompahgre Peak in Colorado down to 350 m on Lake Mead Reservoir. The UCRB produces 90% of the river's discharge [27], with nearly half (seasonally varying) of UCRB baseflow being derived from groundwater [11,22]. Overall, the UCRB serves as an exploitation/extraction-dominated basin, with geologic, forest, agricultural, water and hydropower, and recreational resources exported downstream or elsewhere in the Southwest (e.g., [28]). The 344,440 km$^2$ Lower Colorado River Basin (LCRB) lies in the Basin and Range geologic province, a late Cenozoic, tectonically extensional terrain dominated by horst and graben mountain ranges [29]. The weather of the CRB is continental and arid, with a strongly bimodal precipitation pattern, particularly in the LCRB [30,31].

Lying between the two geologic provinces, the Grand Canyon ecoregion (GCE) occupies 35,000 km$^2$ of the CRB that drains into Grand Canyon, Arizona's world-renowned, deeply incised canyon [32]. The GCE extends from lower Lake Powell Reservoir 500 km downstream to Lake Mead reservoir. Grand Canyon, as well as other national, state, and county parks, and wildlife preserves, refuges, and wilderness areas contain many ecologically intact springs in the CRB. Those springs can serve as relatively undisturbed sentinel sites for understanding aridland springs ecosystem and conservation ecology (e.g., [8,16,33,34]). We also conducted an intensive investigation of the discharge, springs, and SDT in the Verde and Virgin River Basins in north-central Arizona and southwestern Utah, respectively. Both of those basins span the geologic province boundary and are threatened by groundwater appropriation [33,35–39].

Most of the land area in the CRB is managed by federal agencies, particularly the U.S. Forest Service, the U.S. Bureau of Land Management, the U.S. National Park Service, and 29 Native American Tribes. There are >50 major national and state parks, monuments, and wildlife refuges in the CRB, including Rocky Mountain, Canyonlands, Grand Canyon, Bryce, and Zion National Parks, as well as many wilderness and recreation areas, which collectively attract more than 10 million visitors per year. Water allocation in the CRB was politically divided by the 1922 Colorado River Compact into upper and lower basins, the former including the states of Colorado, New Mexico, Utah, and Wyoming, and the latter including the states of Arizona, California, and Nevada (Figure 1) [17,40]. The U.S. Bureau of Reclamation manages 15 major federal dams, and all of the Colorado River's flow is lost to abstraction, infiltration, and evapotranspiration before it reaches the United States–Mexico border [10–15,22,27]. Additional climatological, geographic, hydrogeologic, socio-cultural, and natural resource management information on the CRB is available (e.g., [12,13,26,38,41,42]) and in the Supplementary Materials. Cognizant of the above basin definitions and issues, we discuss springs in relation to both the CRB's hydrologic and its political boundaries; however, we do not include here geographic consideration of groundwater and springs in landscapes outside of the CRB that are influenced by the Colorado River, including eastern Colorado, the Great Salt Lake Basin in Utah, and southern California.

## 3. Springs and Associated Species Data Sources

We compiled information on 6755 partial or full inventories of springs in the CRB states between 1985–2019 (e.g., [4,8,25,42–46]) using standardized geographic (including georeferencing and habitat area measurements), discharge, water characteristics, vegetation, zoology, and human impact protocols [45]. We compiled data on SDT from those inventories, and from museum data-mining at most of the southwestern university collections and the U.S. National Museum of Natural History, interviews with taxonomists, and from the literature. Literature on springs-dependence, rare species, and species of management concern is widely scattered and is found among databases available from each CRB state, regional, and federally listed species lists, as well as individual species accounts (e.g., [45]). Inventory data and information on taxonomy, distribution, life history, endemism, and springs-dependence were compiled and entered into Springs Online (springsdata.org; see Supplementary Materials).

We scored endemism on the basis of the number of populations (usually individual springs) at which a species occurred, from many across a continental range (a score of 0) to occurrence at only a single springs ecosystem (a score of 6) [25]. We scored springs-dependence on the basis of the extent to which springs were required for the species existence, from entirely accidental occurrence at springs (a score of 0) to species for which the entire life history takes place at the springs source (a score of 6). Springs-dependence varied somewhat between plants and animals due to potential facultative occurrence of plants, and thus plants were scored separately from animals.

## 4. Human Demography Data and Analyses

We defined an initial CRB boundary using the standard watershed delineation tools in ArcGIS Pro 2.4.1 [47] on the basis of 30 m digital elevation model data from the National Elevation Dataset [48–50]. Along most of the boundary, this watershed delineation closely aligned with National Watershed Boundary 8-digit hydrologic unit code (HUC) edges [48,50]. For comparability with other HUC-based analyses, we defined our final analysis area to include all of the 8-digit HUCs whose centroid occurred within our initial basin boundary polygon. Due to boundary similarities, we used a combination of CRB hydrological and political boundaries to divide the UCRB from the LCRB [17,40]. We used custom Visual Basic for Applications (VBA) functions in ArcMap 10.5.1 [51,52] to further subdivide the CRB into 100 m elevation bands and areas within 5 m of National Hydrography Dataset perennial streams [53].

Within each of these subdivided polygons, we calculated the planimetric ($x$, $y$) area, surface area, and surface area ratio by applying geodesic methods directly onto the curved surface of the planet [54,55]. To calculate basin elevation statistics in relation to landscape configuration, water characteristics, and human population trends, we generated a systematic array of points across the basin at a density of 1 point per 7.71 km$^2$ ($N = 81,332$ points), and extracted the elevation at each point using bilinear interpolation on the digital elevation model, as well as human population density in 1990 and 2010 [25].

We wrote custom VBA functions to calculate the total number of springs per polygon, as well as the number of spring types based on Springer and Stevens' spheres of discharge typology [56]: cave, exposure, fountain, geyser, gushet, hanging garden, helocrene, hillslope, hypocrene, limnocrene, mound-form, and rheocrene. Springs geographic, physical, ecological, and human use data were collected using standardized inventory and assessment protocols [45] compiled from the literature [1,7–9,25,43,57–73], and are archived [25]. We examined elevation patterns and UCRB versus LCRB differences in springs discharge, water characteristics (including the commonly reported field variables of pH, water temperature, and conductivity), habitat area, and human population trends using simple linear regression and density plots. We reduced the skew of discharge and specific conductivity data by $\log_n$ transformation. We summarized springs data by elevation band and sub-basin using custom VBA functions, and analyzed statistical trends using R 3.6.1 [74].

To determine the 1990 to 2010 human population growth rate and risks to springs and SDT, we calculated human population density at each grid point (above) and at each springs location. To do so, we divided the census block population count by polygon area. We calculated the $\log_n$-transformed triple decade-interval change in $\log_n$-transformed population density data, dividing those values by 20 yr for the temporal linear regression analysis [62–64]. We analyzed temporal patterns in population density change at springs locations and by springs type using one-way ANOVA tests, and checked for statistically significant pair-wise differences in elevation between spring types, as well as for Tukey's honestly significant difference (HSD) tests. This test adjusts $P$-values to control for type I error among large numbers of pairwise comparisons [74].

ANOVA tests assume that experimental errors are normally distributed and that counts and variances among groups are equal. A Shapiro–Wilks test for normality revealed a statistically significant departure from normality in our data ($P < 0.01$), and thus a potential violation of these assumptions; however, visual examination of histogram data demonstrated that this departure was minor, and that

statistical significance was due to the large sample size, and was not biologically or demographically meaningful. Bartlett tests for heteroscedasticity revealed that mound-form spring type had a higher variance than other types, and that hanging gardens had lower variance, whereas other springs types appeared to be statistically homoscedastic. We report only those results that were significant at $P < 0.05$.

## 5. Results and Discussion

### 5.1. Springs Distribution

We found reports of 20,872 springs distributed throughout the CRB, of which data from 6755 inventories confirmed the presence of at least 3386 (16.2%) springs (Table 1, Figure 2). Among reported springs, 8052 springs are in the UCRB, and 12,820 springs are in the LCRB. CRB springs exist widely across elevation, from below sea level in the Salton and Salado basins, to >4000 m in the Rocky Mountains. Springs are patchily and strongly non-randomly distributed in relation to stratigraphy and geologic structure, emerging in and around escarpments; in volcanic and horst-and-graben mountain ranges; and, more rarely, on desert valley floors. For example, the southwestern quarter of Arizona supports an extremely low density of springs, whereas the southern Colorado Plateau in north-central Arizona contains many of that state's largest springs. Throughout the CRB, large springs emerging at low elevations or in flatlands tend to be well known and named, whereas smaller springs at high elevations or emerging in topographically complex landscapes are not. Some springs types tend to go unrecognized as springs, particularly helocrene wet meadows and hypocrene buried springs [56].

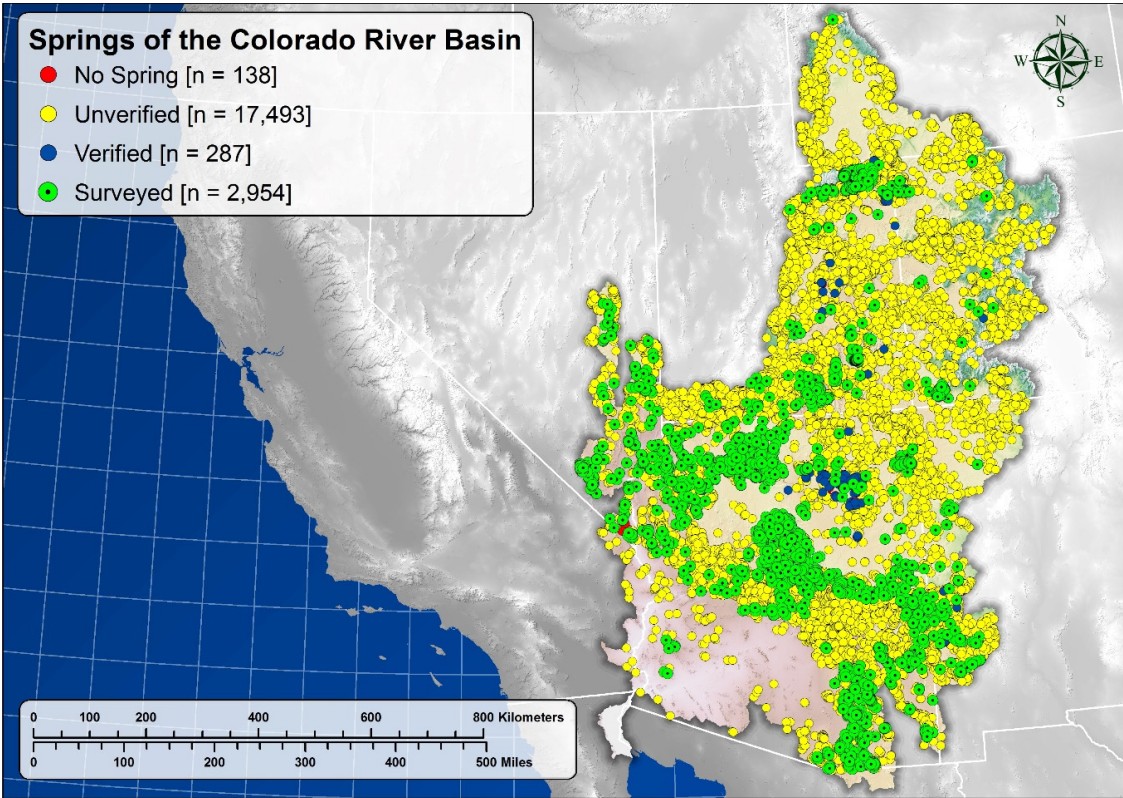

**Figure 2.** Map of the reported and inventoried springs of the Colorado River Basin. Surveyed springs are green; verified but not inventoried springs are dark blue; and unverified, un-inventoried springs are yellow. Springs mapped erroneously are red.

**Table 1.** Average geographic, physical, and geochemical characteristics of Colorado River Basin springs, by sub-basin. Simple linear regression equations (SLR; with F statistics) are provided for springs discharge, and water temperature and geochemistry variables across elevation (Elev; m). Asterisks (*) indicate statistical significance ($P \leq 0.05$).

| Variable | Upper CRB Springs | | Lower CRB Springs | | Overall CRB Springs | |
|---|---|---|---|---|---|---|
| | Descriptive Statistics | SLR Elev Model | Descriptive Statistics | SLR Elev Model | Descriptive Statistics | SLR Elev Model |
| Planview surface basin area (km$^2$) | 283,384 | — | 344,440 | — | 627,823 | — |
| Number of reported springs | 8052 | — | 12,820 | — | 20,872 | — |
| Mean springs area (m$^2$), estimated total springs habitat area ± 95% CI (km$^2$) (% of total land area) * | 3700, 29.9 ± 9.8 (0.011%) | — | 1200, 15.5 ± 3.76 (0.005%) | — | 2200, 45.4 ± 4.4 (0.007%) | — |
| Perennial stream length (km) | 25,821 | — | 64,351 | — | 90,172 | — |
| Springs discharge + 95% CI L/s (median; N) * | 0.74 ± 0.44 (0.07; N = 211) | Y = −3.92 + 0.001X ($F_{1205}$ = 6.86, $R^2$ = 0.03, P = 0.009) | 0.10 ± 0.48 (0.11; N = 1476) | Y = −0.41 − 0.001X ($F_{1,1362}$ = 46.6, $R^2$ = 0.03, P < 0.0001) | 0.09 ± 0.04 (0.10; N = 1687) | Y = −0.61 − 0.001X ($F_{1,1569}$ = 47.4, $R^2$ = 0.03, P < 0.0001) |
| Mean water temperature ± 95% CI (°C) (median; N) * | 13.7 ± 0.77 (13.5; N = 281) | Y = 22.53 − 0.004X ($F_{1279}$ = 54.8, $R^2$ = 0.16, P < 0.0001) | 17.1 ± 0.4 (16.4; N = 1364) | Y = 26.28 − 0.005X ($F_{1,1362}$ = 428, $R^2$ = 0.24, P < 0.0001) | 16.5 ± 0.3 (15.9; N = 1645) | Y = 25.87 − 0.005X ($F_{1,1643}$ = 557.0, $R^2$ = 0.25, P < 0.0001) |
| Mean pH ± 95% CI (median; N) * | 7.19 ± 0.09 (7.35; N = 356) | Y = 9.01 − 0.001X ($F_{1354}$ = 165, $R^2$ = 0.32, P < 0.0001) | 7.47 ± 0.04 (7.50; N = 1335) | Y = 7.85 − 0.0002X ($F_{1,1333}$ = 56.1, $R^2$ = 0.04, P <0.0001) | 7.41 ± 0.04 (7.47; N = 1691) | Y = 8.08 − 0.0004X ($F_{1,1689}$ = 205, $R^2$ = 0.11, P < 0.0001) |
| Mean electrical conductivity (µS/cm) ± 95% CI (median; N) * | 616.1 ± 153.8 (263.5; N = 334) | Y = 10.0 − 0.002X ($F_{1332}$ = 404, $R^2$ = 0.55, P < 0.0001) | 642.7 ± 59.9 (414.56; N = 1292) | Y = 7.33 − 0.001 ($F_{1,1290}$ = 360, $R^2$ = 0.22, P < 0.0001) | 637.2 ± 57.0 (398; N = 1626) | Y = 7.72 − 0.001X ($F_{1,1624}$ = 781, $R^2$ = 0.32, P < 0.0001) |

Thus, CRB springs mapping data are of varying quality, and accurate mapping remains an important limitation. We report at least 138 erroneously mapped springs (cases in which no springs existed at or near a reported location) among reported springs. This indicates an over-estimate of 0.7% among reported springs. However, many springs throughout the CRB remain undocumented, the result of varied inventory intensity among states. Intensive CRB inventories have documented moderate to high levels of previously unreported springs (e.g., [8,21]). For example, Ledbetter et al. [43] discovered 7 (9.9%) previously unreported springs among 71 sites they visited. Reducing the total estimated number of springs by the percent of erroneously reported springs, and estimating that at least 9.9% of springs remain to be discovered suggests that the CRB many contain as many as 22,800 springs.

The distribution of reported springs across elevation differs from that of the landscape itself. The density of UCRB springs peaks at a distinctly higher elevation (2400 m) than does that of land area (2000 m), likely due to the rarity of springs in flat-lying plateau lands and the abundance of montane GDE fens [43,75]. The low density of springs relative to land area at elevations above 3400 m likely result from limited land area, as well as reduced mapping intensity in those often inaccessible locations. In contrast, LCRB springs density peaks slightly lower than the landscape in general (1500 m vs. 1750 m). However, a more interesting difference between the two sub-basins is the large proportion of LCRB land area occurring below 1000 m but supporting relatively few springs (Table 1; Figure 2).

*5.2. Habitat Area*

As elsewhere, most springs in the CRB are small—among 320 inventoried UCRB springs, the mean springs-influenced habitat area is 3700 m$^2$ (95% CI = 1200 m$^2$), a value greatly influenced by the relatively high abundance of large, high-elevation GDE fens (Table 1) [76,77]. Among 755 inventoried LCRB springs, the mean habitat area was 1200 m$^2$ (95% CI = 300 m$^2$), and skewed by the large size of helocrenic fens and ciénegas.

There are few estimates of springs habitat area elsewhere. Springer et al. [76] reported a highly variable average springs area of 682 m$^2$/spring among 56 southern Alberta springs, a much smaller value than we report, and closer to our median UCRB habitat area of 436 m$^2$. Similarly, a subset of 81 springs in Kaibab National Forest's North Kaibab Ranger District in northern Arizona had a highly variable mean area of 742 m$^2$ (95% CI = 406 m$^2$), ranging from 4.5 to 15,036 m$^2$ [43]. Again, that value was closer to our median value rather than the mean, suggesting that the preponderance of high elevation wet meadow fens in the UCRB likely skewed our mean value. Additional inventory is needed to refine these area estimates, but on the basis of mean areas, we estimated the total springs habitat area to be no more than 29.9 km$^2$ in the UCRB, 15.5 km$^2$ in the LCRB, and 45.4 km$^2$ in the overall CRB, making up 0.004% to 0.011% of the total land area in each sub-basin, or 0.007% of the overall basin land area.

*5.3. Springs Types*

Twelve primary springs types [56,77,78] exist in the CRB, but the frequency of different types varies greatly, as follows (Figure 3):

Rheocrene > Hillslope > Helocrene ≈ Hanging Garden >> Cave > Hypocrene > Gushet ≈ Mound-form ≈ Exposure > Limnocrene > Fountain > Geyser.

The rarest natural springs types in the CRB are geysers and fountains, with only one to a few of each presently known, respectively. Anthropogenic alteration or creation of all springs types occurs, including many springs excavated to form limnocrenic livestock watering "tanks", and including the single anthropogenic "coke bottle" geyser (Crystal Geyser) near Green River, Utah [79].

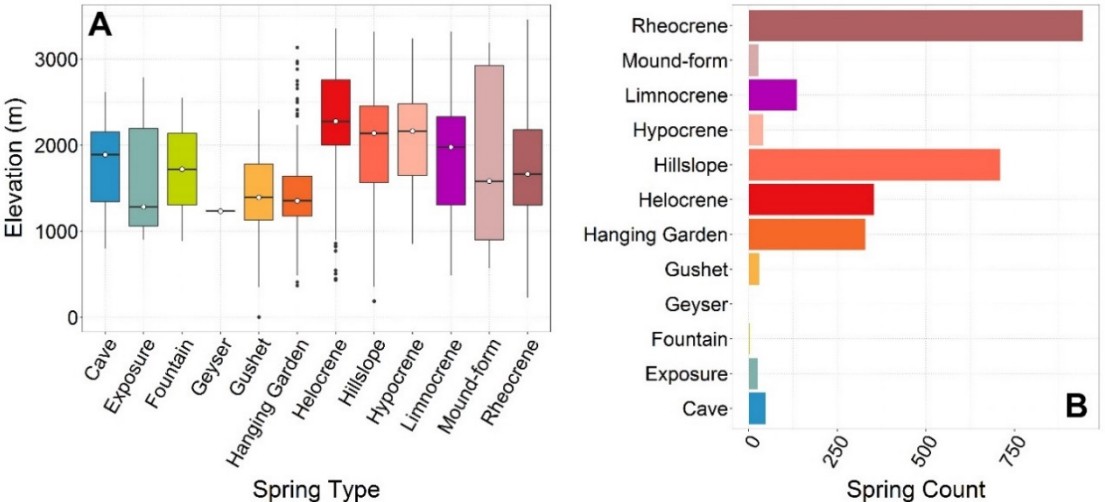

**Figure 3.** (**A**) Distribution and (**B**) count of springs types across elevation in the Colorado River Basin, calculated on the basis of inventoried springs.

The frequency of springs type varies greatly between the sub-basins, altering the above dominance patterns in the UCRB versus the LCRB (Figure 3). More helocrenes and hanging gardens occur in the UCRB, but hanging gardens and geysers are rare or non-existent in the LCRB. We found significant differences among springs types across elevation (one-way ANOVA $F_{1,11} = 45.44$, $p < 0.0001$, Tukey's HSD test $P < 0.05$). For example, helocrene springs are the dominant UCRB springs type at high elevations, primarily occurring as montane GDE fens, whereas low-elevation ciénegas, which are desert GDE fen equivalents, are rare and are highly threatened by anthropogenic activities throughout the LCRB [65,75,80].

Several other CRB springs types warrant special mention. Southern Utah and the GCE are underlain by large sandstone aquifers that produce many hanging gardens springs [81–84]. Hanging gardens develop highly distinctive alcove and wet backwall geomorphic microhabitats. Another pattern is that limnocrene and travertine mound-forming pool springs are rare on the UCRB Colorado Plateau, but are somewhat more common in the Sonoran and Mohave Deserts in the LCRB [85].

*5.4. Flow*

Average CRB springs discharge varies little across elevation, although discharge was found to be slightly positively related to elevation in the UCRB (Table 1) [85]. Springs provide a substantial quantity of baseflow to UCRB perennial streams [11], a contribution that is proportionally increasing with the loss of permanent snowfields and glaciers at highest elevations (Figure 4a) (e.g., [24]). However, in Arizona, springs discharge differs between the two geologic provinces [85,86]. Springs on the Colorado Plateau portion of Arizona contribute 84% of the state's overall springs flow, with the greatest amounts of discharge coming from the Grand Canyon region, the southern edge of the Colorado Plateau, and the Sky Islands of southeastern Arizona. The highest discharge value recorded in Arizona was from the Blue Springs complex and associated springs in the lower Little Colorado River drainage (mean discharge = 5663 L/s). Although derived from multiple springs (not just a single source), Blue Springs discharge is a significant outlier, being 3.3-fold greater than the second highest discharge (Havasu Springs, 1700 L/s; the single largest springs discharge value) [85].

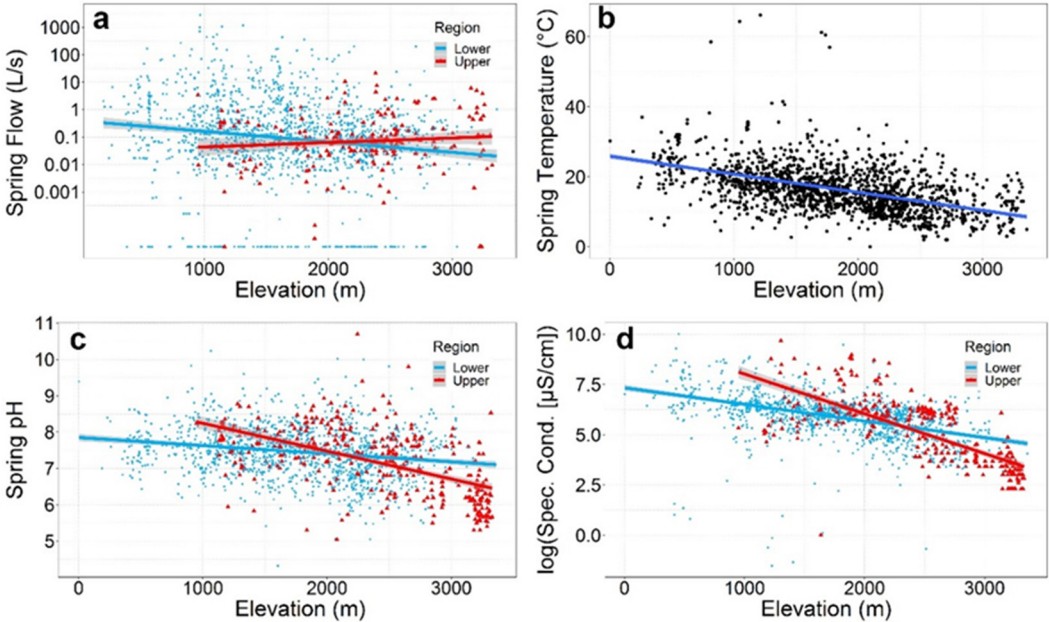

**Figure 4.** Springs discharge and water characteristics in relation to elevation in the Colorado River Basin: (**a**) springs discharge (L/s); (**b**) water temperature (°C); (**c**) pH; and (**d**) specific conductivity (µS/cm).

All of the nearly 50 natural perennial streams in Grand Canyon and all natural streams in Arizona are baseflow sourced by springs [16,21]. However, although only approximately 100 L/s of springs water is discharged on the North and South Rim plateau lands, springs within Grand Canyon on both sides of the Colorado River contribute approximately 10,000 L/s to the mainstream, with longer residence times among south side springs [43,67,71,87,88].

Springs provide perennial baseflow for streams throughout at least the LCRB. In the Verde River basin, many perennial tributary sources have not been accurately mapped. However, field excursions and the literature reveal that all of the known perennial reaches of the Verde River basin were baseflow fed by large helocrenic (e.g., Del Rio Springs), rheocrenic sources primarily emanating from the karstic Redwall Formation (e.g., Fossil Springs in Fossil Creek; Parsons Springs in Sycamore Creek), or springs emerging from sandstone or shallow basaltic aquifers [36,86], along with small hillslope and some alluvial sources. Although poorly documented, the perennial baseflows of the Verde and Virgin Rivers appear to be nearly entirely springfed [21,33,38,39].

Arizona springs are sufficiently well known to estimate their total contribution to the state's water supply. Extrapolating from the mean discharge of 1342 recently measured sources, and adding 5221 L/s of springs-contributed baseflow from major Colorado River perennial tributaries, springs contribute at least 17,176 L/s of water (0.54 km$^3$/year), the equivalent of 18% of the state's annual Colorado River Compact allotment [85].

Discharge variation within individual springs is generally poorly monitored in the CRB, but differs by aquifer and springs type [66,87]. Recharge is primarily derived from snowmelt, and discharge variation is greatest in shallow aquifer and karstic systems (e.g., [59,67,88–90]). Shallow aquifer volcanic aquifers at middle and upper elevations generally respond strongly to climate, and many go dry in years with low snowpack—approximately one-third of the springs on the southern Colorado Plateau appear to be ephemeral. The Grand Canyon ecoregion is largely karstic, with stacked Paleozoic and Mesozoic aquifers. The former produce large, coolwater springs that commonly undergo one to two orders of magnitude in annual discharge variation [67,86,90]. Although hanging garden springs often occur in small sandstone recharge basins, they similarly are susceptible to climate variation [84]. Springs that are least susceptible to climate change are those that emerge from deeper aquifers with longer groundwater residence time (sometimes exceeding $10^4$ yr), greater mineral enrichment, and warmer temperature. Such springs are largely decoupled from surface processes [91,92].

Thus, uncertainty prevails over the extent of, and variation in springs discharge across the CRB, particularly because of the limited quality of mapping and monitoring data. Many springbrooks throughout the CRB are "losing streams", with emerging groundwater flowing some distance before sinking back into their channel floors. Consequently, much springs discharge is not included in surface flow computations at streamflow gauges. The UCRB contains many high elevation springs that have not been mapped, and few discharge measurements are available. In the LCRB, many formerly large springs have not been measured in recent decades, and may have been dewatered. For example, multi-decadal monitoring of Del Rio Spring in central Arizona (the source of the Verde River) reveals that discharge decreased from 60 to <14 L/s from 1997 to 2018 due to groundwater pumping [25], and is now nearly dry. Similar decreases in springs discharge have occurred elsewhere in Arizona [93], including in active groundwater management areas, due to groundwater withdrawal.

*5.5. Water Temperature and Geochemistry*

CRB springs water temperature and geochemistry vary widely among springs in relation to aquifer, elevation, and groundwater residence time and flowpath (Figure 4) (e.g., [43,59,67,87,92]); however, and unlike discharge among shallow-aquifer and karstic springs, water temperature and geochemistry generally varies relatively little within springs over time. Natural water geochemistry can be mineralized, particularly in deep-aquifer or magmatic, long-flowpath springs, such as Kendall Warm Springs in southwestern Wyoming [94], Moapa Warm and other springs in southern Nevada [95,96], and Fossil and Verde Hot Springs and Montezuma Well in central Arizona [38,97]. However, naturally unpalatable waters often support diverse, locally endemic life, including hydrobiid snails, naucorid water bugs, and endangered fish, which entirely rely on specific geochemical conditions for survival (e.g., [97–100]).

Springs water temperature is strongly inversely related to elevation throughout the CRB (e.g., [43]), ranging from near 0 °C to > 66 °C (Table 1, Figure 4b). Springs emerging from basalt aquifers at high elevations often have relatively short flowpaths and typically produce cool water with low ion concentration. In contrast, springs with long flowpaths often produce warmer, or more rarely, hot, highly mineralized and sometimes travertine-depositing water. Travertine-depositing springs emerge primarily at lower and middle elevations in intermountain valleys and in geothermal settings (e.g., Montezuma Well in central Arizona, Blue Springs in the lower Little Colorado River, Havasu Springs in lower Grand Canyon, Fossil Creek in the Verde Valley in Arizona; [71,86–88,91,96]).

Springs water temperature is warmer and slightly less variable in the LCRB as compared to the UCRB (Table 1). Geothermal springs in the UCRB occur at scattered locations in the Rocky Mountains (e.g., Glenwood Springs in Colorado), and in the lower Virgin River basin in Utah and throughout the LCRB (e.g., Pah Tempe Hot Springs in Utah; Arizona, Castle, and Verde Hot Springs in Arizona). The hottest springs (*n* = 69 springs > 30 °C) occur in the central portion of the LCRB, along the inter-provincial, tectonic boundary that extends from southern Nevada along the southern edge of the Colorado Plateau into central New Mexico. Although naturally arising, such highly mineralized waters generally are neither palatable nor safely potable. However, greater diversity in water temperature leads to greater niche diversity and the opportunity for evolution of endemism among taxa, such as the Gila Springsnail (Hydrobiidae: *Pyrgulopsis gilae*) [98]. Most CRB geothermal springs have lengthy histories of modification for recreational use, including many that have been used as resorts; consequently, few undisturbed sites exist at which to study geothermal springs ecohydrology.

The pH of CRB springs is inversely related to elevation (Table 1, Figure 4c). The median pH of UCRB springs is 7.19 (95% CI = 0.09), increasing to 7.47 (95% CI = 0.04) in the LCCRB. Springs pH values are elevated in karstic landscapes and carbonate basins, decreasing at high elevations (e.g., [43]). We report that 32% of UCRB variation in pH is attributable to elevation, related to differences in aquifers between sub-basins. A steeper negative slope in pH exists cross elevation in the UCRB as compared to the LCRB, attributable to the greater abundance of acidic montane fens and bogs, with mats of decaying moss and other wetland vegetation [75,101]. Extreme pH values occur in mined

or other highly altered landscapes. Decreased soil development (e.g., [102]) and plant assemblage richness in low elevation ciénegas (below), as compared to montane helocrenic GDE fens may be at least partially attributable to groundwater and springs pH differences.

The general pattern of electrical conductivity (EC) follows that of pH, with an inverse relationship between EC and elevation, and with variation in relation to aquifer characteristics (Table 1, Figure 4d; e.g., [43]). UCRB median EC is 616 μS/cm (95% CI = 153.8 μS/cm), increasing to 642.7 μS/cm (59.9 μS/cm) in the LCRB. Similar to pH also, the slope of UCRB EC in relation to elevation is steeper than that in the LCRB. These patterns are related, in large part, to the relatively brief residence time of groundwater in upper elevation igneous aquifers, and to the generally longer residence time in LCBR deep aquifers.

## 5.6. Springs-Dependent Taxa

*Overview:* SDT biodiversity, distribution, and habitat requirements in the CRB have not previously been comprehensively investigated, and a full list of species remains outstanding (Table 2). Existing information is scattered through the literature, and inventories and ecological or conservation studies are nearly all focused on individual taxa or within land units or state boundaries. Nonetheless, such inventories regularly document the presence of endemic, rare, or disjunct populations, and sometimes report previously unrecognized taxa (e.g., [103]). Below, we present information on >330 CRB SDT, including invertebrates, vertebrates, and plants [25], and we describe several emergent biological patterns and additional information, including references on each SDT in the Supplementary Materials.

**Table 2.** Estimated and known springs-dependent taxa in the Colorado River basin. Estimates of Ephemeroptera, Plecoptera, and Trichoptera springs-dependent taxa (SDT) richness are based on the percent of known species with restricted elevation ranges.

| Taxon | Estimated Number of SDT | % of Total SDT |
|---|---|---|
| Plants | 62 | 18.8 |
| Mollusca | 21 | 6.4 |
| Non-Insecta Arthropoda | 7 | 2.1 |
| Insecta—Ephemeroptera | 21 | 6.4 |
| Insecta—Odonata | 37 | 11.2 |
| Insecta—Hemiptera | 22 | 6.7 |
| Insecta—Plecoptera | 30 | 9.1 |
| Insecta—Coleoptera | 22 | 6.7 |
| Insecta—Trichoptera | 47 | 14.2 |
| Insecta—Lepidoptera | 9 | 2.7 |
| Chordata—Fish | 33 | 10.0 |
| Chordata—Amphibia | 9 | 2.7 |
| Chordata—Repilomorphs | 4 | 1.2 |
| Chordata—Aves | 1 | 0.3 |
| Chordata—Mammalia | 5 | 1.5 |
| Plants | 62 | 18.8 |
| Invertebrates | 216 | 65.5 |
| Vertebrates | 52 | 15.8 |
| Total | 330 | 100.0 |

*Microbial Ecology:* The microbial ecology of springs is a vast field of inquiry into past and contemporary biodiversity, and the advent of environmental DNA methods is providing profound insight into microbial assemblage complexity [104]. Although much research has focused on hot springs and submarine vent systems, recent research is exploring microbial influences on mineral precipitation at springs. Microbes and biofilms can facilitate and be fossilized in mineral deposits, and scanning electron microscopy analysis of tufa deposits at Cold Springs, Utah, provides evidence of paleosprings microbial life as far back as the Jurassic Period, in characteristic landforms that may be identifiable on other planets [105]. Besides health-related research on topics such as the lethal

"brain-eating" parasitic Excavata amoeboflagellate, *Naegleria fowleri* [106], at Arizona and other CRB hot springs, research on microbial ecology is just beginning in the CRB.

*Worm Phyla:* Nematoda, Platyhelminthes, and Annelida worms abound in and around CRB springs, but there has been relatively little taxonomic investigation of those phyla. Neophoran geoplanoidean freshwater flatworms are common in near-neutral, cool to ambient springs at middle and upper elevations. Although Turbellaria are not generally regarded as indicators of high water quality, they often are found in association with Plecoptera. However, flatworm diversity has not been systematically studied in the CRB, and the fauna is likely to be far more diverse than is presently recognized. Among the clitellatan acanthobdellidan Annelida, *Motobdella montezuma* and at least two *Helobdella* leeches (Erpobdellidae) are endemic [97,107]. Gordian horsehair worms (Aschelminthes: Nematomorpha, Gordiacea) are common in lentic CRB springfed settings, attesting to the high productivity and attractiveness of their hosts to springs habitats.

*Mollusca:* The CRB supports at least 21 SDT aquatic and springs riparian Mollusca, including 17 locally endemic trucatelloidean springsnails. Locally endemic hydrobiid springsnail species (particularly *Pyrgulopsis*) are diverse and threatened in the Basin and Range province, but are comparatively depauperate on the Colorado Plateau [98,108]. Another aquatic endemic SDT gastropod is *Physa zionis* (Physidae) [109]. In addition, many terrestrial wetland land snail species are SDT, such as the succineid genus *Oxyloma*. Spamer and Bogan [110] reported high species richness of land snails around Grand Canyon springs, and North [69] reported densities of small terrestrial land snails in low desert springs exceeding $10^4/m^2$; however, the latter study reported no evidence of springs-dependence or endemism among the many small-bodied terrestrial gastropods. In addition to gastropods, but far less well known taxonomically, finger clams (Sphaeriidae: *Pisidium*) commonly occur in CRB springs and springfed streams.

*Chelicerata and Myriapoda (Arthropoda):* Aquatic and riparian SDT spiders are dominant predators at CRB springs (e.g., *Dolomodes triton* fisher spider, *Tetragnatha* long-jawed spiders, and many lycosid wolf spiders). Aquatic mite diversity also is likely high, but poorly studied in the CRB. CRB Myriapoda habitat affinities also are poorly known, but *Tyloborus utahensis* (Spirobolidae) is a rare, disjunct SDT riparian millipede reported from Dutton Springs in central Grand Canyon [111].

*Crustacea: Hyallela* amphipods (Gammaridae) are ecologically important and abundant in CRB springs (e.g., [97]), and many undescribed taxa may exist. *Brachichecta kaibabensis* fairy shrimp are endemic in GDE ponds on Grand Canyon's North Rim [112]. On the basis of springs studies in northern Mexico, many SDT micro-crustaceans also are likely to occur at CRB springs, but few data are available. We note that the occurrence of Ostracoda and Cladocera at CRB springs is typically associated with anthropogenic habitat alteration.

Although aquifer-dependent rather than springs-dependent, subterranean freshwater amphipods in the genus *Stygobromus* are diverse in aquifers along the Pacific Coast of North America, but the genus is depauperate in the CRB. Wang and Holsinger [113] reported only 8 of 53 (15.1%) western North American *Stygobromus* species in the CRB, of which *S. fontinalis*, *S. simplex*, *S. utahensis*, and *S. wardi* occur in the UCRB; *S. arizonensis*, *S. blinni*, and *S. boultoni* occur in the LCRB; and *S. herbsti* occurs in the upper White River drainage in Nevada. Of those, *S. arizonensis*, *S. fontinalis*, and *S. herbsti* have been reported exclusively from springs, and *S. blinni* is endemic in Roaring Springs Cave in Grand Canyon. The relatively low diversity of CRB *Stygobromus* may be partially the result of incomplete sampling, but is mirrored by the depauperate biodiversity of CRB truncatelloidaen springsnails [98]. Due to their subterranean habitat affinity, we did not include *Stygobromus* in our tally of SDT (Table 2).

*Ephemeroptera:* Although at least 104 Ephemeroptera species exist in the CRB [114], species distribution and habitat data are generally poorly known. We found that at least 21 (20.1%) of CRB mayfly species have highly restricted elevation ranges, suggesting that they may be SDT. Among recognized SDT, *Moribaetis mimbresaurus* (Baetidae) is endemic to the springfed headwaters of Oak Creek in central Arizona, and is the only member of its genus in North America [115]. At best a crude estimate, considerably more research is needed into CRB mayfly habitat requirements.

*Odonata:* Ninety species of Odonata have been detected in the Grand Canyon ecoregion, including several SDT [116]. Several Anisoptera species are SDT, including one *Aeshna*, two *Cordulegaster*, and one *Brechmorhoga* dragonflies. Many damselfly species exist primarily in springs and low-order springbrooks, but *Coenagrion resolutum* (Coenagrionidae) is restricted to high elevation springs and springfed ponds in northern Arizona and the UCRB.

*Aquatic Hemiptera:* Stevens and Polhemus [117] identified 89 species of aquatic or semi-aquatic Hemiptera (ASH) in the Grand Canyon region, of which more than one quarter appear to be SDT. For example, 6 of 10 Naucoridae, including species such as *Ambrysus c. circumcinctus*, have only been detected in springs or springfed streams. Out of the total ASH fauna reported, nine species are SDT Mexican/Neotropical species with highly disjunct distributions. For example, *Micracanthia quadrimaculata* (Saldidae) occurs at a single, isolated alkaline springs complex in central Grand Canyon. *Ochterus rotundus* (Ochteridae), and several other isolated ASH populations similarly indicate LCRB habitat connectivity to Mexican neotropical landscapes over geologic time.

*Plecoptera:* Stonefly diversity in the UCRB in Colorado and Utah includes at least 94 species ([118,119]; L.E. Stevens, unpublished data). Among SDT taxa, widespread *Malenka coloradensis* (Nemouridae) typically is associated with high elevation seeps and springfed streams [120], whereas *Malenka murvoshi* has been reported only from two springs in the Spring Mountains of Nevada [121]. Although Plecoptera diversity is highest at upper elevations in the UCRB, SDT Plecoptera also occur at low–middle elevations in the LCRB. For example, *Anacroneuria wipukupa* Redrock Stonefly occupies springs and low-order springfed streams at elevations down to about 1000 m in north-central Arizona [122]. We estimate that at least 20% of Plecoptera species in the Intermountain West of North America are SDT. Using the proportion of species with narrow elevation ranges as a surrogate for SDT status yields 30 CRB species (31.9%), an only slightly higher value than our estimate. Again, this crude estimate will be modified as understanding of Plecoptera habitat use and distribution improves.

*Coleoptera:* Several families of aquatic beetles and many individual species are nearly or entirely restricted to springs in the CRB. For example, Dryopidae (long-toed riffle beetles, three genera), Elmidae (riffle beetles; eight genera), and Psephenidae (*Psephenus murvoshi* (Psephenidae; trout stream beetles) [123] are recognized SDT. Some species in those families are more widespread among springs and low-order springfed streams in the UCRB, but are more tightly restricted to one or a few springs in the LCRB. Even widespread stream habitat generalists, such as *Microcylloepus similis* (Elmidae) would be largely eliminated by the loss of springs habitats. Among the predaceous diving beetles, many *Agabus*, *Laccophilus*, and other genera are widespread in North America [124], but in many cases the only localities at which they have been detected are at springs. The large, well-known *Cybister ellipticus* (Dytiscidae), which occurs from California and across the southern USA, is known in the entire LCRB only from a single limnocrene in lower Grand Canyon (L. Stevens, unpublished data). Similarly, some haliplid and hydrophilid aquatic beetles (e.g., some *Berosus*) are virtually only encountered at CRB springs. Blinn [97] noted that *Anacaena signaticollis*, *Crenitulus* nr. *debilis*, *Laccobius ellipticus*, and *Enochrus sharpia* hydrophilid beetles are known in Arizona and the USA only from Montezuma Well. Riparian, as well as aquatic beetles are recognized as springs endemics. For example, the Grand Canyon Wetsalts Tiger Beetle (*Cicindelidia haemorrhagica arizonae*) occurs only along the wet margins of low-order perennial springfed streams in central Grand Canyon [125], occupying a total estimated habitat area of no more than a few hectares [16]. Similarly, although not strictly SDT, other carabid ground beetles (e.g., *Bembidion* spp., *Nebria* spp., *Omophron* spp.), as well as Heteroceridae and other beetle taxa, are most often found along springbrooks, but habitat affinity on most species remains outstanding. Alarmingly, springs-dependent Stephan's Riffle Beetle (*Heterelmis stephani*) was endemic to a few springs in the Santa Catalina Mountains near Tucson, but recently was reported as having passed into extinction [126], a fate that threatens many CRB SDT.

*Diptera*: A similarly large, but even more poorly known true fly fauna occurs at CRB springs, but no rigorous systematic inventory has been conducted. Among the Nematocera, many Tipulidae and Chironomidae [127], but no Culicidae appear to be exclusively SDT [128]. Among the Brachycera,

several families appear to include some to many SDT, including Tabanidae, Dolichopodidae, Empididae, and Stratiomyidae. For example, Spence [68] conducted an inventory of springs in Glen and Grand Canyons, from which four new species of empidid dance flies were discovered. In the Acalyptratae, Mathis and Mathis [129] reported at least 68 species of Ephydridae shore flies in Grand Staircase-Escalante National Monument in southern Utah, with several species new to science, and many of which were SDT.

*Trichoptera:* Thus far we have documented 380 caddisfly species in the CRB ([130,131]; L.E. Stevens, unpublished data). As with other aquatic invertebrate taxa, specific knowledge of the extent of springs-dependence is poorly known for most species; however, multiple species within several families appear to be springs-dependent. These include one or several species in each of the families of Apataniidae (e.g., *Apatania arizona*), Hydroptilidae (e.g., *Metrichia* spp.), Limnephilidae (e.g., *Limnephilus frijole*), as well as members of the families Odontoceridae, Phryganeidae, Philopotamidae, and Polycentropodidae. Again using the proportion of species known to have restricted elevation ranges, we estimate that at least 47 CRB Trichoptera species (12.3%) are SDT.

*Lepidoptera:* Butterflies occur in 2- to 2.5-fold order of magnitude greater abundance, and 3- to 5-fold greater species richness at northern Arizona springs as compared to adjacent uplands [8]. In addition to water and salt, some of this intensive occurrence may be related to consistently high soil moisture, which often is required for pupal survivorship. Among the noteworthy SDT butterflies in the region are Nokomis Fritillary (*Speyeria nokomis*) and Four-spotted Skipperling (*Piruna polingii*), which are considered by the US Forest Service as sensitive species along the Mogollon Rim. SDT moth species in the CRB include at least five species of aquatic *Petrophila* (Crambidae) in spring-fed Oak Creek, Arizona, of which four are newly described [132].

*Vertebrates:* Among vertebrates, springs dependence is most prevalent among fish. Although depauperate overall, the CRB is renowned for having among the highest proportion of endemic fish species in North America, with at least 35 subspecies among 33 endemic species [18]. Of that total, 23 (65.7%) are partially or entirely SDT, particularly including small desert fish species. Among many examples of the latter are cyprinodontid *Cyprinodon* (e.g., *C. macularius* Desert Pupfish [133,134], goodeid *Empetrichthys*, cyprinid *Moapa* and *Rhinichthys*, and poeciliid *Poeciliopsis* topminnows. At least two native salmonids (e.g., *Oncorhynchus apache* and *Oncorhynchus gilae*) are restricted to low order, spring-fed LCRB streams, and the largest breeding population of cyprinid *Gila cypha* Humpback Chub, a large big-river cyprinid, spawns primarily in the outflow of Blue Springs in the lower Little Colorado River [16]. Extinction threatens many SDT fish, and some (e.g., the Monkey Springs Pupfish, *Cyprinodon arcuatus*) have already been lost [135].

CRB helocrenic and springbrook SDT herpetofaunae include Sonoran Tiger Salamander (*Ambystoma mavortium stebbinsi*), at least four *Lithobates* leopard frogs, several *Anaxyrus* toads, and at least two federally listed aquatic garter snakes (*Thamnophis* spp.) [136–138]. Several springs-dependent mammal species occur in the CRB, including soricid shrews and *Zapus hudsonicus* jumping mouse [139,140]. *Microtus* voles are meadow taxa, but in the LCRB voles are commonly found around springs, and springs may serve as population refugia during drought [8]. Southwestern American Dipper (Cinclidae: *Cinclus mexicanus unicolor*) nest behind or adjacent to madicolous, springfed waterfalls, building their dome nests of moss e.g., [141]. Although many bird species rely on springs for water (e.g., [8]), populations of cinclids appear to be the only truly SDT birds.

*Plants:* CRB springs phycological biodiversity studies are few, although at least three endemic diatoms have been reported from Montezuma Well [97]. Similarly, the composition of CRB springs-dependent bryophytes has yet to be studied. In contrast, CRB vascular plant and fern distribution has received robust attention, revealing at least 59 SDT among the >4000 species in the CRB (Table 2, SI). Notable endemic SDT vascular plants include Rolland's bulrush (*Trichophorum pumilum)* and Wolf's orache (*Atriplex wolfii* var *tenuissima*) at seeps and springs in western Colorado and southwestern Wyoming [142]; canyonlands and Navajo sedges (*Carex curatorum* and *C. specuicola*, respectively) at hanging gardens in the Four Corners area [68]; McDougall's flaveria (*Flaveria mcdougallii*) in alkali

springs in central Grand Canyon [10,143]; Arizona and Bebb's willow (*Salix arizonica* in Arizona's White Mountain GDE fens, and *Salix bebbiana* throughout the high elevation springbrooks and helocrenes); and Canelo Hills ladies' tresses orchid (*Spiranthes delitescens*) and Huachuca water umbel (*Lilaeopsis schaffneriana* ssp. *recurva*) in southeastern Arizona [144]. In addition, we detected several extremely rare or singular populations of springs-dependent *Buddleja sessiliflora* and *Eryngium sparganophyllum* during LCRB Arizona ciénega inventories. Several of those species are regarded as rare [145], and several have been considered for, or have received federal protection.

*5.7. SDT Patterns*

Our list of CRB SDT is preliminary and continues to grow as we discover additional information, but several patterns appear from the data presently available. Springs often support assemblages containing an array of aquatic, wetland, and riparian taxa (of which some may be rare or locally endemic), as well as widespread upland species, which often occur facultatively around the springs periphery (e.g., [60,66,69,108]). Assemblage mixing creates high levels of species density (species packing), often more than an order of magnitude greater at springs than in the surrounding uplands [8]. For example, inventories of 81 North Kaibab Ranger District springs in northern Arizona documented 441 plant species, one-third of the total forest vascular plant species richness (1325 species) on 6.01 ha of inventoried habitat, 0.001% of the overall landscape [43]. Although improved understanding of plant species–area relationships is needed, CRB springs support a disproportionally high percentage of the regional flora. Therefore improved stewardship of springs may greatly enhance regional conservation of plant biodiversity.

The CRB offers excellent opportunities to understand the potential impacts of climate change through space-for-time analyses across its >4 km elevation gradient. One facet of such research is the relationship between elevation on the frequency and rarity of CRB wetland plant and animal SDT. Analyses of inventory data, floras, and state rare species lists indicates asymmetry in the extent of endemism between springs-dependent vascular plants and animals. A total of 142 plant taxa are recognized as rare in Arizona by the Arizona Rare Plant Commission [145], of which only 22 species (15.5%) are wetland or aquatic taxa, and only 5 (3.5%; two *Carex*) are SDT. However, the state list of rare species is incomplete. A recent catalogue of Arizona *Carex* (Cyperaceae) [146] and discussion with the authors of that study revealed that 5 of 68 (7%) Arizona *Carex* species are considered by those experts to be rare (only two had previously been included in the state rare plant list), but that 25 of 68 (37%) *Carex* species are SDT, including two taxa that are new to science.

The composition and diversity of CRB wetland *Carex*, *Juncus* (Juncaceae), and other wetland plant taxa decreases with elevation below 2500 m. This pattern warrants more attention, but may be attributable to both evolutionary and physiological factors [66,147,148]. Many of those species are Nearctic in distribution, and may exist at upper elevations as Pleistocene relictual assemblages. In addition, the inverse elevation and pH relationship at CRB springs (above) results from the prevalence of acidic peat-forming GDE fens in high elevation catchments with crystalline bedrock geology, and the contrasting high alkalinity in low elevation springs with longer groundwater residence times. Low elevation aquifer geochemistry appears to present a selective challenge that is too harsh for many Nearctic wetland plant taxa.

At low elevations in the LCRB, Felger [149] reported 514 plant species in the Gran Desierto at the mouth of the Colorado River in northwestern Sonora. However, only three (<0.6%) species were even remotely springs-dependent—locally rare but otherwise widespread *Lythrum californicum* and *Eleocharis rostellata* were encountered at Quitobaquito and La Salina springs, and endemic *Distichlis palmeri* occurred widely on saline wetlands, but was not solely restricted to GDE habitats. Additionally, small artesian springs ("pozos") emerge in and around saline playas on the western edge of Bahía Adair at the head of the Sea of Cortez [150]. That field of pozos supported 27 wetland and halophilic wetland plant species, nearly all of which are widespread and none of which are SDT. Thus, despite the relatively high frequency of endemic (particularly dune-dwelling) upland plant species in the

Gran Desierto, few wetland and no truly springs-dependent endemic vascular plant species occur there. These data lend further support to the basin-wide observation that helocrene (wet meadow, fen, ciénega) plant species richness and diversity, including springs-dependent and endemic wetland species, is positively related to elevation, with lowest values at springs near sea level.

Although our studies have added additional rare vascular plant SDT to the CRB tally, asymmetry in the frequency of endemism between plants and animals remains strikingly apparent. In contrast to the 59 CRB springs-dependent vascular plant species we report (Table 2), at least 216 SDT invertebrates and 35 CRB fish taxa (nearly two-thirds of CRB fauna) are SDT. Many of both animal groups are endemic to, restricted to, or spawn only in springs and low-order springfed streams. A similar discrepancy between floral and faunal SDT endemism exists in Nevada [7].

The relative paucity of endemic SDT vascular plant species may be attributable to several factors. First, the pattern may be partially an artifact of insufficient data, as in the case of the Arizona *Carex* [146]. Second, the pattern may be attributable to insufficient attention to description of habitat data. For example, the use of the habitat descriptors "springs", "seeps", or "wet meadows" is erratic in published floras. Improved information quality will certainly increase the number of recognized SDT plant species and clarify the role of springs in plant rarity; however, it is unlikely to much increase the number of already well-studied endemic plant species, as will certainly continue to occur among aquatic invertebrate taxa (e.g., [108,151]). The pattern may be partially attributable to the order-of-magnitude greater richness of SDT invertebrate taxa, and also may be related to plant biology. Many wetland plant species are pollinated through anemophily or by zoophily by widespread pollinator taxa, allowing maintenance of gene flow even at remote sites and precluding the evolution of endemism. In contrast, aquatic faunae often become closely adapted to distinctive water temperature and geochemistry and habitat conditions, conditions that enforce isolation and limit gene flow.

Endemic plant species that occur at springs often appear to be relictual populations, wetland habitat specialists that disappear as springs are dewatered by direct anthropogenic actions or by climate change. Such species are characteristic of hanging gardens, which support several locally endemic plant species (e.g., *Carex spicuola*, *Primula specuicola*, *Cirsium rydbergii*), and a highly distinctive plant assemblage [66]. Given the usually small groundwater catchments of those springs, it is remarkable that the endemic populations remain extant. In contrast, many endemic faunal species appear to be adaptational endemics, closely affiliated to the habitat conditions of their particular springs and unable to exist elsewhere. Thus, evolutionary differences between floristic relictualization and faunal adaptation also may be responsible for asymmetrical endemism among SDT.

Much progress has been made in understanding the diversity and ecology of CRB SDT, but many enticing research topics remain to be explored. Many SDT microbes and invertebrates have yet to be discovered, identified, and described, and much remains to be learned about the basic taxonomy, habitat requirements, biogeography, and conservation status of a great many SDT. Unfortunately, many species face imminent threats of extirpation and extinction due to habitat loss, and a high percentage of recent extinctions in the CRB are of SDT (e.g., [98,126,133,135,145]).

*5.8. Socio-Cultural Significance and Demographic Impacts*

Native American cultures in the arid CRB have long used and revered springs as important hunting, gathering, and agriculture sites, and as places of spiritual and historic significance ([4] and references therein). Indigenous Uto-Aztecan (Pueblan), Yuman, and several Shoshonean and Athabascan linguistic groups in the CRB regard individual springs as portals of cultural emergence and return into different dimensions, sites of ceremonial and initiation, sites that host divine beings, sites with healing power, and waypoints along prehistoric and historic migration routes [152]. Some of these perspectives were appropriated and adopted to some extent by the Hispanic culture that colonized the region beginning in the 16th century, beliefs that became secularized and filtered into contemporary Anglo-European culture, and in part into western USA appropriative water policy and law, as well as the springs water bottling industry [12,152,153].

In 2017, the resident CRB human population was at least 11.3 million people, including 2.1 million in the UCRB and 9.2 million in the LCRB [154]. This does not include more than 10 million tourists per year who visit the many natural wonders and anthropogenic attractions of the basin. Nonetheless, 4.4-fold more LCRB residents are downstream recipients of UCRB water supplies. UCRB water also is exported to the Front Range of Colorado, and LCRB water is exported to southern California and Mexico watersheds [12]. Collectively, those urban areas supported 24.7 million people in 2017, a population 2.2-fold greater than that in the entire CRB [154]. Thus, LCRB and extra-basin human population demands on the famously over-allocated waters of the CRB exceed the available supply. Unfortunately, ineffective state-based groundwater policy and state–federal surface water policy jointly create abundant opportunities for conflict over CRB water supplies, and limited prospects for resolution.

Our analyses corroborate these demographic patterns, with several-fold greater population density in the LCRB as compared to the UCRB. However, average population density at springs is much lower at than away from springs, both at the scale of the whole basin and within sub-basins (Table 3, Figure 5). A total of 39% of the CRB landscape had no population in 2010 and no population change from 1990 to 2010. We attribute this to the extent of federal land management throughout the basin, which precludes private residence. Consequently, 35% of the pixels holding springs had no human occupation. Population density generally increased at spring locations in all regions, but at much lower rates than within the basin as a whole. This pattern is not unexpected because springs water rights often were established early in the 20th century, and few new opportunities exist for population expansion at springs. Thus, population increases at springs are minor and statistically insignificant. However, regionally increasing population densities affect springs by increasing local and regional pumping, diversion, and other human activities.

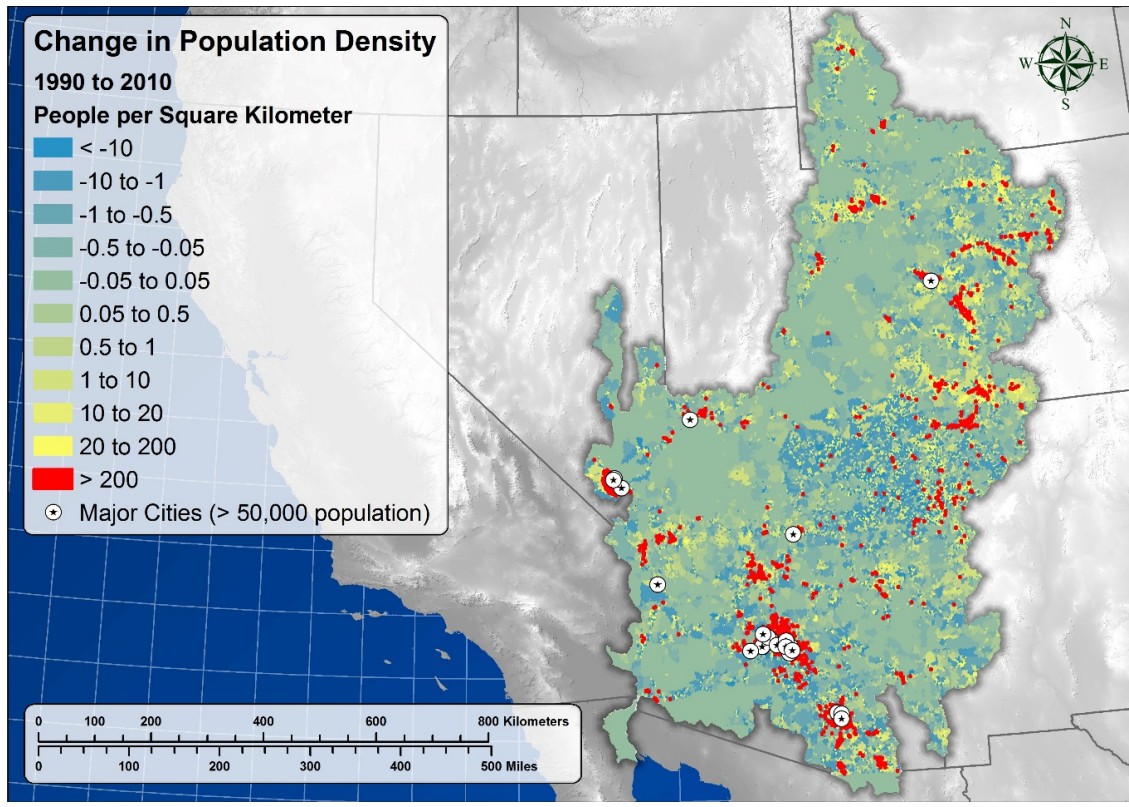

**Figure 5.** Human population density (residents/km$^2$) from 1990 to 2010 in the Colorado River Basin.

**Table 3.** Human population density (people/km$^2$) and change in population by sub-basin and in the Colorado River Basin overall, and in the vicinity of springs from 1990 to 2010. Total population estimates in the sub-basins and the overall basin are provided in the text. Paired between-year and inter-sub-basin comparisons are significantly different at *P* < 0.05.

| Basin Name | Within Entire Basin | | | At Spring Locations | | |
|---|---|---|---|---|---|---|
| | Census Block 1990 Density | Census Block 2010 Density | Census Block Mean Annual Change in Density 1990–2010 | Spring Count | Mean Spring 2010 Density | Mean Annual Change in Spring Density 1990–2010 ± 95% CI |
| Upper Basin | 2.02 | 3.07 | 0.05 | 8052 | 3.36 ± 1.89 | 0.11 ± 0.06 |
| Lower Basin | 11.36 | 22.92 | 0.58 | 12,820 | 6.7 ± 3.24 | 0.12 ± 0.11 |
| Full Basin | 7.26 | 14.00 | 0.34 | 20,872 | 5.41 ± 2.10 | 0.12 ± 0.07 |

The LCRB human population doubled between 1990 and 2010, and increased by 1.5-fold in the UCRB during that time period. Levels of anthropogenic impairment of inventoried springs exceeds 70% in many CRB landscapes (e.g., [8,43]), and exceeds 90% where populations are increasing most rapidly, particularly in desert valley landscapes, such as Las Vegas, Nevada, and in Arizona's Groundwater Active Management Areas [155]. Direct localized impacts include those at- and post-emergence factors, such as source water extraction, piping, livestock watering, construction, non-native species introduction, and recreational use. Although local impacts can be resolved if the supporting aquifer is relatively intact [85], policy changes typically are needed to resolve regional stressors [155]. Our inventories throughout the CRB indicate that localized springs impairment commonly is related to under-informed livestock management practices; however, urbanization and diffuse settlement now threaten many springs throughout the developing portions of the CRB, and deep-aquifer impacts may accrue through accelerating intensive oil and gas hydraulic fracturing operations in the UCRB [156].

Groundwater pumping is intensifying, particularly during drought [70], and is predicted to transition springs geomorphology and ecosystem ecology from their natural typology and condition to hyporheic alternate states [157]. When the water table drops below the groundwater-supported phreatic and rooting zone of wetland and riparian plant species, upland vegetation comes to dominance at the former springs. This process is a form of hydrosere (hydrarch) succession, although the ecology of this process has not been well-described at aridlands springs.

Human introduction of non-native plants, invertebrates, and vertebrates also poses major threats to springs ecosystems and SDT by altering disturbance, competition, predation, parasitism, and disease regimes [99,100,158]. Many non-native plant species occur at CRB springs, including aquatic *Chara* and other algae; *Agrostis*, *Arundo*, *Bromus, Eragrostis*, *Phalaris*, and other grasses; a plethora of exotic herb species; and Saltcedar (*Tamarix* spp.), *Ulmus*, Russian olive (*Eleagnus angustifolia*), as well as *Washingtonia* palms [158]. These non-native plants alter soil, nutrient, light, and water availability and increase fire frequency, and the woody species can dewater smaller springs.

Non-native plant occurrence at springs is common but varies across elevation. Relatively few non-native plants occur at highest elevations, such as the Rocky Mountain alpine tundra and springs, whereas non-native weeds can be strongly dominant at middle and upper desert elevations. However, and non-intuitively, the relationship between native and non-native species is often positive—springs and riparian habitats with high native richness are likely those with high non-native richness [1,34,43,158]. Slight natural or anthropogenic habitat disturbance opens niche space in these often highly productive habitats, facilitating weed invasion. Felger [149] reported a total of 75 non-native (17.0%) species in the Gran Desierto, a percentage of non-native wetland plant species (11 of 72 species, 15.3%) equivalent to that of the upland flora (64 of 442 species, 14.5%). This suggests that at lowest CRB elevations, wetlands (including springs) are not necessarily more susceptible to non-native plant invasion than are upland habitats, a pattern in marked contrast to the greater frequency of non-native species in wetland and riparian habitats in middle and upper CRB elevations (e.g., [1,158]).

Among non-native animals, introduced crayfish (particularly *Procambarus* and *Orconectes* spp.), New Zealand mudsnails (*Potamopyrgus antipodarum*), red-rimmed melania snails (*Melanoides tuberculata*), quagga mussel (*Dresseina bugensis*), *Corbicula* clams, and other Bivalvia have become widely established in the Lake Powell, the lower Colorado River, as well as low-elevation LCRB tributaries, and can colonize springs. Non-native crayfish and aquarium and game fish exert strong negative impacts on rare native and endemic springs-dependent aquatic invertebrates, fish, and amphibians (e.g., [99,100]). *Lernaea* copepods and at least 15 other fish parasites have been spread by carp (Cyprinidae: *Cyprinus carpio*) and more than two dozen other game and aquarium fish species throughout the CRB [16]. *Batrachochytrium* fungus has been associated with introduced American bullfrogs (*Lithobates catesbiana*), and is regarded as a significant threat to native amphibian populations, particularly in warmer regions [159]. Terrestrial non-native vertebrates include feral equines and livestock that alter habitat

structure, soil, vegetation, and nutrient availability, and feral pets such as house cats also can threaten SDT. One unique advantage of middle and lower elevation CRB springs is that their springbrooks often are "losing streams", aquatic habitats that are disconnected from perennial stream networks, protecting the springs from upstream invasion by aquatic non-native animals.

## 6. CRB Springs Rehabilitation

On a positive note, mindful groundwater and springs habitat stewardship can reverse dewatering, habitat degradation, and hydroseral succession, and SDT populations can be rehabilitated if the aquifer is relatively functional, conserved, recovering, or is restored (e.g., [34,35,160]). Local springs stewardship practices that have facilitated CRB springs rehabilitation often involve simple, inexpensive measures. Ecologically sensitive CRB springs assessment, planning, rehabilitation, and development practices have been described for private, non-governmental, state, and federal natural resource stewards (e.g., [4,35,85,155,161,162]). Simple, cost-effective rehabilitation actions include subterranean (pre-emergence) flow-splitting division of the discharge to ensure flow at the biologically important source, appropriate fencing that provides livestock and wildlife with access to water away from the source (but see Kodrick-Brown and Brown [163]), construction of trails to prevent hillslope erosion, and removal of non-native species [34,85,160]. Springs rehabilitation has now become a common theme in state and federal landscape management planning, but review of such stewardship efforts demonstrates the need for monitoring to evaluate continued project success [9,35,75,85,160]. Although invasive species sometimes can be controlled, considerable effort and time may be required for full removal, particularly of rhizomatous phreatophytic plants, crayfish, exotic fish, and bullfrogs (e.g., [34]).

Springs rehabilitation projects on the CRB have successfully regenerated habitat, which can then be used to translocate and recover extirpated species. For example, translocation of *Lithobates onca* relict leopard frogs has proven successful at springs along the shores of western Lake Mead [164]. Habitat restoration also can allow reactivation of buried wetland seed reservoirs. Rehabilitation of Pakoon Ranch Springs in northwestern Arizona allowed rapid germination and population re-establishment of a regionally rare wetland SDT plant, *Eustoma exaltum*, from seeds long-buried in wetland soils there [34]. Thus, improved stewardship can result in positive conservation of SDT, and even reverse apparent extirpation [160].

## 7. Conclusions

The basin-wide scale of our inquiry is relevant because the highly regulated Colorado River is transitioning from a snowmelt-dominated to a groundwater-fed fluvial-riparian ecosystem [12,19,23,165]. Groundwater overdraft, local habitat impacts, and the introduction of non-native species threaten many springs, reducing regional biodiversity, and degrading the CRB landscapes that those springs influence [4,26]. To better plan for this inevitable transition, additional research, planning, and adaptive actions are needed to understand groundwater sustainability, springs and SDT distribution and status, and stewardship options. Such information and considerations may help prevent additional needless habitat and species losses.

Reduced snowpack and surface flows, increased human demands and impacts on aquatic resources, and the continuing introduction of non-native species are collectively generating unheralded changes in CRB ecosystems (e.g., [12,166]). Springs in large, urbanized regions are already largely dewatered or lost, including those near Denver, Las Vegas, Los Angeles, Phoenix, San Diego, and Tucson (e.g., Ojos Caliente County Park in Tucson; Pipe Springs National Monument near Fredonia, Arizona; the wet meadows for which Las Vegas, Nevada, was named; and others). However, in many cases smaller montane and foothills aquifers that produce springs often are little-affected by regional anthropogenic activities except climate change. Those small-aquifer springs are more likely to be threatened by local agricultural, domestic, or recreational uses, and non-native species introductions. Springs sourcing from relatively intact aquifers may continue to function with ecological integrity, and springs have demonstrated remarkable resilience and ecological rehabilitation potential, often with

relatively inexpensive treatment [34,35,85]. Advancing understanding of springs as ecosystems and improving stewardship practices are urgently needed in this large, arid river basin, as it is throughout the world [60].

Effective groundwater management is critical for guaranteeing the sustainability of the natural and sociocultural heritage in the arid CRB, and elsewhere [152]. In addition, improved stewardship of springs will disproportionally amplify conservation of SDT. Water can be extracted from springs for human uses while maintaining sufficient discharge and ecological integrity at springs sources, making springs among the most sustainable natural ecosystems [4,162]. At the local scale, simple best management practices can be used to help balance economic and environmental management (appropriate fencing, maintenance of water conveyance, trail construction, monitoring, etc.). However, regional and climate change threats must be faced at national and international societal scales.

Thus, the protection of springs provides an important avenue for effective point-source biodiversity conservation, and illuminates larger natural resource policy and management issues (e.g. [9]). We established Springs Online (SpringsData.org) as a relational information management portal for documenting; archiving; and analyzing the distribution, characteristics, status, and management of springs to improve scientific understanding and stewardship in the CRB and globally [25,78]. We invite those interested to review and contribute data into this rapidly growing database. As the CRB transitions towards a new and uncertain future [23,166], this vast and diverse basin provides a globally important example of how interactions among aridlands climate, groundwater, surface water, and biodiversity influence and can help sustain human society.

**Supplementary Materials:** The following are available online at http://www.mdpi.com/2073-4441/12/5/1501/s1. We provide additional descriptive geographic information on the study area. We also provide a Microsoft Excel workbook with data on springs water temperature, geochemistry, springs-dependent species and scoring criteria, as well as human population change from 1990–2020, on which analyses presented in the manuscript were based.

**Author Contributions:** L.E.S. contributed manuscript concept and organization, data analyses, writing, review, and correspondence. J.J. provided GIS analyses, mapping, writing, and review. J.D.L. provided data, writing, and review. All authors have read and agreed to the published version of the manuscript.

**Funding:** Although preparation of this manuscript was not funded, some of the diverse research that contributed to the manuscript was supported by the US Bureau of land Management (L15AC00074), Bureau of Reclamation (R18AC00022), the US Forest Service (18-CS-11030700-014), the Grand Canyon River Outfitters' Colorado River Fund, and private contributions to the Museum of Northern Arizona.

**Acknowledgments:** The research behind this manuscript and reported upon here was supported by many individual agencies, organizations, and individuals, and by much-appreciated donations to the 501c3 Museum of Northern Arizona Springs Stewardship Institute (SSI). A. Hazelton, B. Mann, A Mendoza, and T. Schipper of SSI provided much-valued assistance with information management and quality control. The data presented here are publicly available and non-sensitive.

**Conflicts of Interest:** The authors declare no conflict of interest.

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
