# Peer review of "Springs and Springs-Dependent Taxa of the Colorado River Basin, Southwestern North America: Geography, Ecology and Human Impacts"

_water, doi:10.3390/w12051501_

Round 1

Reviewer 1 Report

The paper presents the results of a large-scale study of the springs in the vast Colorado River Basin. Since the majority of CRB is in arid climate the water is scarce in general and springs, especially those close to densely settled areas are under great pressure to water withdrawal and other human impacts. The manuscript is interesting and will draw attention of the scientific audience. Methodology used by the authors seems to be correct, but improvements are required before the final acceptance of the paper for publication. What I miss is the standard structure of the article where authors set some of the hypotheses and use the methods to accept or reject them. I also suggest to shorten some paragraphs and sentences where details and names of several springs are listed, which have the meaning for local audience. The manuscript presents a thorough study of springs and is therefore a valuable contribution. It also contains suggestions for good management practices in order to preserve this natural heritage and water as essential good.

Suggestions for the authors:

Abstract:

Ln 11: I suggest omitting the data about the size of catchment area from the abstract.

Ln 13: I suggest: water-receiving instead water-consuming

Ln 16: writing about baseflow it is enough springs. (without ecosystem). I suggest to change the sentence to: CRB springs are important for biota, culture and economy, but …..

Ln 19:  perennial à permanent

Ln 21: (cliff-dominating) settings à landscape

Ln 25: I suggest replacing geochemistry with the measured parameters pH and conductivity

Ln 31: SDT of plants,….

Ln 34: Endemic SDT plant species richness (62 species) is definitely not low, but high relative to any other group. It is not good idea to compare the species numbers of plants and invertebrates.

Introduction

Ln 49-51: Please rewrite the first sentence.

Please explain the term bio-culturally complex or rewrite this part

Ln 71: Surface area of France is Cca 547.000 km2 , you repeat these numbers in the subchapter Study Area. I suggest to delete here.

Methods

Ln 113:  dominant à the largest and most important

Ln 115: sixt largest river in N.America is not an important fact for this study

Ln 165-169: I suggest to delete this part.

Ln 246: The sentence ends here ?

Ln 249: Table 1. The unit for conductivity is μS/cm.

Ln 263: Please delete this part of sentence.

Ln 331-333: Please delete this part of sentence.

Ln 334: Please delete Figure 4 since it is not displaying springs.  

Ln 348, 349, 373:  Please delete the names in brackets.

Ln 352-355: Please delete these sentences.

Ln 360: Please delete this part of sentence.

Ln 367: Please delete: “ with stacked Jurassic over Permian over”.

Ln 382: Please delete this sentence.

Ln 393-395: Please delete this part

Ln 405-407, 410, 411, 428: Please delete the names in brackets.

Ln 413-415: Please delete this part of sentence.

Ln 423-424: Please delete this sentence.

Ln 435: The unit for conductivity is μS/cm.

Conclusions:

Ln 827-828: Please delete these lines.

Ln 840-841: Please delete the names in brackets.

References cited:

Ln 947, 949: Please check these references.

Reviewer 2 Report

The authors of “Springs and Springs-dependent Taxa of the Colorado River Basin, Southwestern North America: Geography, Ecology and Human Impacts” have prepared a timely, well-written, and impactful review of the natural history and socio-ecological context of springs in the Colorado River Basin, which covers a large portion of the southwestern US. This manuscript, along with the accompanying database described therein, provides a critical snapshot of spring ecology in this region, synthesizing available data and creating a benchmark and reference for future work.

I present some minor suggestions related to presentation below. Beyond these, I only have two suggestions. First, from my perspective, the authors have used appropriate and relevant tools to analyze their geographic, water quality, biodiversity, etc. data. Yet due to the nature of the manuscript, a reader’s confidence in their findings also needs to rely somewhat on the credibility of the authors. Something that might be helpful would be a statement in the Methods authenticating the authors as spring experts drawing on considerable shared experience (as seems to be the case). This might preempt any concerns about some subjectivity in methods.

Second, the review presented here is highly comprehensive, but the penultimate section on rehabilitation comes across as quite brief. Since this section presents an opportunity to highlight what can be done to protect and restore springs, I would love to have read a little more about these opportunities/techniques. If the editors believe that there is space for the authors to expand a bit on rehabilitation, it would be exciting to read more on the topic.

As someone hailing from the LCRB, I am grateful to the authors for putting together such a synoptic and thoughtful review of the region’s springs!

General Stylistic

I am not an expert in the study of springs and defer to the authors to use the most common conventions. But I wonder if it is better to change “springs” to “spring” when it is the first word in a noun phrase. This happens first in lines 15-16 (“springs ecosystems” as opposed to “spring ecosystems”) and then occurs throughout. Unless there is a convention in the spring(s) world to keep the “s,” I would lose it.

Commas in large numbers are sometimes missing (e.g. line 12) and really help when they are present (e.g. line 13).

Abstract

Generally, I wonder if the abstract could be shorter. Its presentation is clear, but some findings are included that don’t seem particularly essential for a reader who is just encountering the abstract. An example of this is the comparison (34-35) of plant endemic to invertebrate endemic richness. While the statement is true and clear, is there any reason that we should expect plant and invertebrate endemic richness to be comparable?

25-26: To me, it doesn’t make sense to say that “Water geochemistry is negatively related to elevation,” as water geochemistry does not in and of itself have a positive or negative valence. In contrast, the next clause, in which pH and conductivity are related to elevation is easier to interpret.

31: SDT needs to be defined here as it is the first time a reader will encounter this term in the manuscript.

38: “human populations” should be “The human population”

Introduction

Great first paragraph (49-70)

73: the first “it is” should be replaced with “as”

76: Should be “The CRB also supports…”

93-94: Should be “We review the existing literature and springs…”

Methods

129: “is” can be removed.

131/135: a formatting issue here has split this sentence.

Fig. 1: I can see the legend for this figure, but not the figure itself.

Springs and Associated Data Sources

187-9: I think I understand what the authors did, but am not sure what “Springs dependence varied somewhat between plants” means. Could this be clarified? Differed between plant species?

Results

Table 1: This table efficiently and clearly presents a lot of data. One thing that could make it better: some standardization of significant figures. There is inconsistency (such as a mean and an SD having different numbers of sig figs). Overall, some sig figs could be rounded up without losing any meaningful information in a way that would visually simplify the table.

Fig. 2: This map looks great. To increase impact, especially for skimming readers, the figure legend could contain brief definitions of the four color categories (e.g., what differentiates a surveyed from a verified spring?).

256-260: I am still a little bit confused about what data sources were compared to assess when springs were mis-mapped or unverified. I know this is the results, but if there is a way to quickly remind the reader how this was determined, it would help.

267: I think you could eliminate this first sentence.

277-292: This is a really interesting analysis. Its reporting could be improved if the authors data (now reported in hectares) and the literature values (now in m^2) were reported in the same units. Given the areas involved, I would suggest m^2 as these are more aligned with the km^2 units used to describe the total spring area of the basins.

299/303: Again, a sentence has been split here.

389, 399, 421, 433: I don’t like these headings at the beginnings of paragraphs. The topic sentence in each case seems to make clear what the paragraph is about. The same goes for the headings in section 5.6.

632-643: Interesting!

Table 3: Same point as Table 1 regarding sig figs.

807/826: Could these be labeled as sections six and seven?

812: What is “splitting the discharge”? Could this be explained in a clause or sentence?

Author Response

This manuscript is a resubmission of an earlier submission. The following is a list of the peer review reports and author responses from that submission.

Round 1

Reviewer 1 Report

The manuscript describes the assessment and (societal) importance of springs (distribution, physical, chemical and biological situation) within the Colorado River Basin. The main objective of the study is framed within the threats caused by changing climate and increasing population. The authors start with an introduction of the basin, followed by a description of the spring types, flows and classification as well as a literature-based assessment of the biotic groups observed within the basin, with specific attention towards spring-dependent taxa. Based on the overall assessment, the authors conclude that better management within the basin is necessary.

From my viewpoint, I believe the study has merit as it highlights the importance of freshwater management on river basin scale , though I feel that there are some opportunities to improve the content and structure of the manuscript in order to transfer the results to the readers in a clearer way. Please find here my comments (no line numbers were available in the file I received):

GENERAL

  • The title covers the content better, yet the subtitle seems to appear on the wrong line;
  • The terminology ‘springs-dependent’ is consistently used throughout the manuscript. Although I would personally prefer ‘spring-dependent’, I leave it to the authors to decide on it.

ABSTRACT

  • Please note that Water allows an abstract of only 200 words;
  • Sometimes, a separation is used in big numbers (e.g. 627,824 versus 627824), though not always;
  • I recommend to place the two main threats/pressures (anthropogenic and climate change) closer together, without being interrupted by a description of the subbasins;
  • “Water quality is negatively related to elevation” can be read as “with increasing elevation, water quality decreases”, which contradicts the majority of literature. Moreover, it is not completely correct to consider ‘water quality’ as temperature, pH and conductance;
  • It is stated that water quality varies little within springs, though this is not discussed in the manuscript, based on the data;
  • Arizona springs contribute 6% to state budget, though I think that the contribution to the overall river flow might be more interesting here.

INTRODUCTION

  • The trophic-dynamic concept of Lindeman was ground-truthed in Florida, though the content of this concept is not discussed. In my opinion, this sentence adds little to the introduction;
  • Paragraph 2: “Climate change is increasing […]”: try combining the things that increase and the things that decrease;
  • Paragraph 3: Mentioning of upper and lower basin, although not introduced before (when not having read the abstract).

METHODS

Study area

  • Paragraph 1: Bracket after range of annual discharge;
  • Paragraph 1: Why is it important to mention the Rio Grande here?
  • Paragraph 2: The geological description seems unnecessary to me;
  • Paragraph 3 (and other): The use of flow units varies throughout the manuscript (km³/year, m³/s, L/s). I recommend using similar units;
  • Paragraph 4: The geological description seems unnecessary to me;
  • Paragraph 6: The Verde river was selected because of its headwaters at 3850 m. What is so special about these conditions that it is selected?
  • Paragraph 8: Lee Ferry is not indicated on the map;
  • Paragraph 9: Bracket after mentioning flow

Springs and associated species

  • Consider numbering this section as 2.2

Human demography

  • Consider numbering this section as 2.3
  • Paragraph 2: Check weblink (http://)

RESULTS

  • Overall, the distinction between literature and own results is not always clear as they are often mixed.

Distribution

  • How is the influence area of a spring defined?
  • Table: The SLR are also depicted graphically later on, consider moving the equations from this table to the graph to obtain a smaller table. Also, mention what CI stands for and split the spring area into two different rows (mean and estimated);
  • Are the F-statistics correct (e.g. I would expect the discharge for UCRB to have F1,209 as statistic instead of F1,205, considering the number of samples);
  • Paragraph 2: 161 mis-mapped springs differs from the 138 mentioned on the map. Also towards the end, the fraction of mis-mapped is said to be 3.3%, which differs from the earlier 4.8%;
  • Paragraph 3: Check ‘USRB’ (also not necessary to repeat this basin here);
  • Paragraph 3: It is not clear how the elevation peaks of spring and landscape are to be derived from Table 1 and Figure 2.

Habitat area

  • Paragraph 2: Discussion on the median UCRB area, while Table 1 reports the mean. I recommend to use only one, or to report both mean and median in the table;

Springs type

  • Figure 3: Consider using vertical layouts for both spring type and spring count and mention the difference between A and B in figure caption. (Also, since all types are explained in the axis, there is no need to use different colours.)
  • Paragraph 1: It is mentioned that limnocrenes are rarest source, though their count (Figure 3B) places them fifth. If this is a consequence of including artificial springs, consider making a distinction between natural and artificial;
  • Paragraph 2: Mentions the difference in distribution and frequency in the subbasins and refers to Figure 3, although this distinction is not visible within this graph;
  • Paragraph 2: The type of ANOVA is not completely clear, was elevation tested across spring types (Figure 3A), or was elevation tested within spring types, but among basins (as part of the distinction between upper and lower basin within this paragraph)?
  • Paragraph 3: Several other springs warrant mentioning, but it is not clear why they deserve specific mentioning.

Flow

  • Paragraph 1: The contribution of springs to baseflow cannot be derived from Figure 4 and 5A;
  • Paragraph 1: Why is there specific focus on the Arizona springs and not on the distinction between the upper and lower basins? The latter would be more uniform with previous sections. Hence, this part does not feel necessary, in my opinion;
  • Paragraph 1: If opting to keep the Arizona springs description, I would recommend to include some flow values, to frame it in the overall flow distribution and contribution of the springs within the CRB;
  • Figure 4: No added value, in my opinion. Can be removed;
  • Figure 5: In Figure 3, capital letters were used for the different subplots, I suggest some uniformity. Moreover, why is there no distinction between upper and lower for temperature? I also suggest to align the figures (e.g., if using R and ggplot2, use the ggpubr package);
  • Paragraph 3: Seems like it fits with paragraph 1, dealing with Arizona springs;
  • Paragraph 5: Starts with mentioning variation, goes to recharge and back to variation;
  • Paragraph 6: If many springbrooks are losing streams, please indicate a percentage of it;

Water Quality

  • It remains tricky to discuss water quality and only mention temperature, pH and specific conductance. Especially since no specific water quality index is calculated nor is it specified what ‘good’ and ‘bad’ physicochemical conditions are;
  • Paragraph 2: Temperature depends on the spring and on flowpath. I initially interpreted flowpath as distance from the spring, but I realised it rather represents the vertical path between aquifer and surface. I recommend clarifying what is meant with flowpath;
  • Paragraph 3: Links temperature with geology and ecology, while I feel that a description on variation would be sufficient here. The statements might be true, but add little to the discussion on the observed temperature range;
  • Paragraph 4: Check the use of LCRB and LCCRB;
  • Paragraph 5: What other aquifer characteristics are meant here and how is that included in Table 1 or Figure 5D)?
  • Paragraph 5: The units to report EC were stated to be µS/cm.

Springs-dependent taxa

  • Overall, there is quite some attention given to the complete CRB, while this is not the focus of this specific work. I recommend to stick to the data in Table 2;
  • Paragraph 1: The statement on the occurring lack of data is not clear from the referenced Table 2;
  • Table 2: Which taxa are known and which ones are estimated? How do the estimates differ from what is known?
  • Table 2: Plants are mentioned twice. Moreover, I recommend to work on two levels (1: Plants – Invertebrates – Vertebrates and 2: Worms – Mollusca - …). In addition, if ‘Other Arthropoda’ is limited to 3, maybe they can be further specified?
  • Paragraph 2: This can be shortly mentioned in previous paragraph, as it was stated that the focus lies on plants, invertebrates and vertebrates;
  • Paragraph 4: The paragraph mentions 12 SDT, while Table 2 mentions 27;
  • Paragraph 17: Personally, I find it difficult to imagine that a bird is completely springs-dependent because of nesting behind waterfalls, as these can also be completely snowmelt-fed. However, this is dependent on how narrow or broad ‘springs-dependence’ is defined;
  • Paragraph 19: Please also specify why better understanding of plant species-area relationships among springs is needed;
  • Paragraph 20: How do the geology and physicochemical patterns effectively explain the observed floristic patterns?
  • Paragraph 25: Unclear sentence: “The pattern may be a function […] taxa than of plants.”

Humans

  • Paragraph 2: The external urban areas accounting for 24.7 million people are completely supported by water from the CRB or also from other sources?
  • Table 3: Not all values have the same units. Recommended to include these values per variable.

CONCLUSIONS

  • Overall, conclusions deal with water and basin management, and limitedly with the performed analyses;
  • Paragraph 2: Not all non-native species are harmful, but almost all change the CRB ecosystem structure and functioning
  • Paragraph 2: No need to specify the lost springs in the conclusions (but, can be considered in the ‘Flow’ section);
  • Paragraph 3: No need to provide management examples (fencing etc.) in conclusions, as these have been introduced before.

The manuscript describes the assessment of springs (distribution, physical, chemical and biological situation) within the Colorado River Basin. The main objective of the study is framed within the observation that changing climate and increasing population cause a rise in the external pressures exerted on the river basin. The authors start with an introduction of the basin and an in-depth characterisation of its hydrology, followed by a description of the spring types, flows and classification as well as a literature-based assessment of the biotic groups observed within the basin, with specific attention towards spring-dependent species. Based on the overall assessment, the authors conclude that better management within the basin is necessary.

From my viewpoint, I believe the study has little merit and marginally aligns with the scope of Water, though I feel that there are a number of opportunities to improve the content and structure of the manuscript in order to transfer the results to the readers in a clearer way. Moreover, I feel that the performed study is more of a review rather than independent research, as it combines data from different sources. The text is mostly a combination of different facts, while the main message tends to be lost by the mixed level of detail and generality.

The setup of the study provides merit as it discusses a complete basin, but the lack of detailed data make an in-depth description challenging for the authors, causing a relatively superficial report on the spring characteristics in the basin. After editing, the manuscript might be considered as a Review (and not an Article).

Reviewer 2 Report

Comments to the article (water 735761):

Springs and Springs-dependent Taxa in the Colorado River Basin, Southwestern North America: Geography, Ecology and Human Impacts submitted by Lawrence E. Stevens, Jeffrey Jenness and Jeri D. Ledbetter

The paper is rewritten and amended version of previous one (water 663357) submitted few months ago.

General Comments:

Ad 1) A topic of the paper exceeds the title of the paper – it includes the whole river and not only springs and spring-dependent habitats. Unless the authors consider the whole river as “spring-dependent ecosystem” – they stated that 50 % of baseflow origins from springs.

Ad 2) One more author is/was added to improve some sections of the MS – I agree

Ad 3) “Capita” of the paper = title + authors + affiliations must be separated (are result of Mac/Windows “translation/communication” problem(s)???)

Ad 4) Citations – new citations are just “dumped” into the text – not follows citations rules (number of references should/must!! appear in order of appearance – see “Instructions for Authors”)   

Ad 5) No detail information is presented how data on sprigs were collected – actually, they can confirm only c. 16% of them by inventory protocols for the rest they collect data elsewhere (but not precisely defined).   

Detail Comments:

  • See attached PDF

General conclusion: Based on all above mentioned comments (both General and Detail), my final decision is again REJECT and submit paper elsewhere. Paper is modified but provide no robust scientific results. In many places includes too many un-necessary details. Could be published in national/regional journal.

Unfortunately I have AGAIN REJECT this paper. It is focused on important and very actual topic, but it is written in too popular and low level style. It is on level of seminar work produced by students (sorry!). Authors made some changes / corrections / modifications, but not enough to be published in Water journal.

There are several technical errors (citation, not well explained methods in M&M section but present in Results; text in figures not clear or even miss-leading) as well as conceptual (it is not clear do the authors are focused on the springs only, or on the whole Colorado river system – from time to time they are focused only on springs, but sometimes are very general).
